# Retrieval of Bulk Hygroscopicity From PurpleAir PM2.5 Sensor Measurements

Jillian Psotka[1,*], Emily Tracey[1,*], and Robert Sica[1,*]

[1]Department of Physics and Astronomy, The University of Western Ontario, London, Canada
[*]These authors contributed equally to this work.

**Correspondence:** Robert Sica (sica@uwo.ca)

**Abstract.**

PurpleAir sensors offer a unique opportunity for a large-scale and densely populated array of sensors to study surface air quality. While the PurpleAir sensors are inexpensive and abundant, they must be corrected to better agree with validated coincident measurements from more sophisticated instrumentation. Traditionally, this correction is performed using statistical methods. We propose a method to both correct the Purple Air $PM_{2.5}$ measurements and allow an estimate of the hygroscopic growth of aerosols, using a novel correction approach based on the optimal estimation method (OEM). The hygroscopic growth of the aerosols can be retrieved using the sensitivity of the correction to water activity, which influences the measured size distribution of the aerosols. By employing the physically-based correction using calibrated measurements from a nearby Ontario Ministry of the Environment, Conservation and Parks air quality site, the average daily Mean Absolute Error (MAE) of the PurpleAir $PM_{2.5}$ measurements is decreased from $5.58\,\mu\mathrm{g}\,\mathrm{m}^{-3}$ to $1.68\,\mu\mathrm{g}\,\mathrm{m}^{-3}$, and the average daily bias decreases from $4.75\,\mu\mathrm{g}\,\mathrm{m}^{-3}$ to $-0.23\,\mu\mathrm{g}\,\mathrm{m}^{-3}$. This improvement in the correct is comparable to that seen using conventional statistical methodologies. Our OEM retrieval also allowed us to estimate seasonal bulk hygroscopicity values ranging from 0.33 to 0.40. These values are consistent with the accepted ranges of bulk hygroscopicity for atmospheric particulate matter (0.1 to 0.9) determined in previous studies using calibrated air quality measurement instruments, which suggests that our method allows a new aerosol product to be determined from a large sensor network.

## 1 Introduction

Home air quality monitors are becoming increasingly popular both for the general public and in the atmospheric science community due to their low cost and ease of use. One notable inexpensive air quality sensor is the PurpleAir PA-II sensor, which can be utilized both indoors and outdoors and is priced at approximately $300 USD. These sensors connect to WiFi and provide users with real-time air quality and ambient condition readings at two-minute resolution. Each PurpleAir sensor can be integrated with the PurpleAir Network, where data from over 30,000 sensors across the globe are publicly available (https://map.purpleair.com, last access; 27 February 2025).

PurpleAir sensors employ light scattering photometry to estimate the concentration of particulates based on an assumed particle composition. Every two minutes, an air sample is drawn through the instrument and the optical laser beam interacts

25 with the particulates in the sample. The scattered light is measured by a photocell detector plate, which converts the detected photons to a measurement of the number of particles in the sample and their sizes. The Plantower PMS5003 sensor used in the PurpleAir sensors is described in more detail in Ardon-Dryer et al. (2020) and Ouimette et al. (2022).

There are two main differences in sensor function that account for the low cost of the PurpleAir sensors relative to research-grade air quality sensors. The first is that the PurpleAir instrument does not have the technology to measure the mass of
30 the particles directly through beta attenuation, therefore it must make assumptions and estimations in order to calculate the photometric particle count (measured in particles per decilitre) and the mass density. These corrections are done onboard the unit using a proprietary algorithm, which introduces additional uncertainty into the data product. The same PurpleAir sensor could read too high or too low in different locations depending on the particle density in that area and how it compares to the assumptions made by the manufacturer. Hence, a correction factor must be applied to the measurements to account for the
35 unknown difference in average particle density.

Additionally, PurpleAir sensors do not have a mechanism to correct for ambient humidity. Humidity can greatly affect the accuracy of PM measurements because particulates experience an increase in diameter in the presence of water (hygroscopic growth). Hygroscopic growth introduces uncertainties in low-cost sensor measurements because they detect higher scattering leading to estimations of higher concentrations. The extent of hygroscopic growth depends not only on the ambient humidity,
40 but also on the aerosol composition. Each type of particulate has its own hygroscopicity, which is a measure of its ability to absorb water. Therefore, a correction for hygroscopic growth must consider $PM_{2.5}$ composition, which varies spatially and seasonally.

PurpleAir sensors have been successfully applied not only to home-use but for scientific use. Bi et al. (2020) incorporated PurpleAir measurements into large-scale $PM_{2.5}$ modelling. They compared 54 PurpleAir sensors to nearby U.S. Environ-
45 mental Protection Agency Air Quality System (AQS) stations across California. They corrected the PurpleAir sensors using a generalized additive model that included linear corrections for relative humidity, temperature, and sensor operating time. The corrected PurpleAir data were then combined with the AQS data to create high-resolution daily $PM_{2.5}$ estimates. In their model, the contribution of data from each individual PurpleAir sensor was down-weighted depending on the residual errors. They found that the model that incorporated PurpleAir measurements was more effective at modelling $PM_{2.5}$ predictions than
50 a strictly AQS-based model. This study is promising in terms of wider applications of PurpleAir data use and could have been improved further with a more accurate PurpleAir correction.

Barkjohn et al. (2021) developed statistical corrections using 50 PurpleAir sensors at 39 unique sites in the United States. They tested 15 linear and multi-linear models of varying complexity, with a mixture of additive and multiplicative interaction terms. They used only parameters that were provided or could be explicitly calculated from PurpleAir measurements so as to
55 make their correction applicable to any PurpleAir site, regardless of proximity to reference instruments. Their study found that PurpleAir sensors' over estimations of $PM_{2.5}$ readings could be adequately corrected by a multiple linear model of the form

$$\text{PM}_{2.5,\text{corr}} = a\text{PM}_{2.5,\text{meas}} + b\text{H} + c, \tag{1}$$

where $a$, $b$, and $c$ are constant coefficients and $H$ is the relative humidity as measured by the sensor. Increasing the complexity of the model did not have significant advantages. They were successful in creating a single nationwide correction that could be applied to PurpleAir sensors and reduce errors. This work was expanded on in Barkjohn et al. (2022) where a correction was developed in cases of extreme smoke concentrations (>300 $\mu$g m$^{-3}$).

Other studies (Ardon-Dryer et al., 2020; Magi et al., 2019; Malings et al., 2019; Tryner et al., 2020; Nilson et al., 2022) have also applied statistical methods to correct the PurpleAir sensors to standard research-grade instruments. In this study, we used an inverse modeling technique called the Optimal Estimation Method (OEM) to correct the PurpleAir measurements using a Thermo Scientific SHARP (Synchronized Hybrid Ambient Real-time) Model 5030 particulate monitor and a physical model of hygroscopic growth factor given by Malings et al. (2019). OEM is explained in detail by Rodgers (2000) and we give a brief description in Section 2.3.

The hygroscopicity of bulk aerosol was also retrieved during the correction process, providing a possible advantage of this technique over previous correction methods. Hygroscopicity is a fundamental parameter describing the ability of aerosol particles to absorb water (Kreidenweis and Asa-Awuku, 2013; Tang et al., 2016). It is important to measure hygroscopicity because it impacts the ability of aerosols to act as cloud condensation nuclei (CCN), thus it affects the formation and properties of clouds and their indirect radiative forcing (Farmer et al., 2015; Petters and Kreidenweis, 2007; Reutter et al., 2009; Su et al., 2010; McFiggans et al., 2006).

## 2 Methodology

### 2.1 The Physical Model

The forward model we used with our OEM method is based on the physics-based correction model used by Malings et al. (2019). This study used two different correction methods, one physics-based and one statistical, to ensure the PurpleAir data would better match regulatory-grade data for nine PurpleAir sensors. The physical model was based on the hygroscopic growth of different aerosols and the composition of the air pollution in Pittsburgh, Pennsylvania (USA). The hygroscopic growth factor of PM$_{2.5}$ which quantifies the hygroscopic growth of the aerosols is given by

$$\mathrm{f}(T,H) = 1 + \kappa_{\mathrm{bulk}} \frac{w(T,H)}{1 - w(T,H)}, \tag{2}$$

where $T$ and $H$ are temperature and relative humidity, $\kappa_{\mathrm{bulk}}$ is the hygroscopicity of bulk aerosol, and $w$ is the water activity. The hygroscopicity of bulk aerosol was calculated as the sum of the fractional component, $x_i$, of each main aerosol multiplied by its hygroscopicity, $\kappa_i$:

$$\kappa_{\mathrm{bulk}} = \sum_{i=1}^{n} x_i \kappa_i. \tag{3}$$

Malings et al. (2019) used four main aerosols, carbonaceous mass, sulfate, nitrate, and ammonium. The fractional composition of these aerosols varies by location, but these are consistently the most abundant PM$_{2.5}$ aerosols in the United States (Bell

et al., 2007). The water activity was calculated as a function of temperature and relative humidity as

$$w(T, H) = H \exp\left(\frac{4\sigma_w M_w}{\rho_w R T D_p}\right)^{-1}, \tag{4}$$

where $\sigma_w, M_w$, and $\rho_w$ are the surface tension, molecular weight, and density of water, $R$ is the ideal gas constant, and $D_p$ is the average particle diameter. A linear correction was also applied to account for the unknown factory calibration of PurpleAir sensors. The total correction was as follows:

$$\mathrm{PM_{2.5,corr}} = a\left(\frac{\mathrm{PM_{2.5,meas}}}{\mathrm{f}(T, H)}\right) + b, \tag{5}$$

where $a$ and $b$ are constant coefficients, $\mathrm{PM_{2.5,corr}}$ and $\mathrm{PM_{2.5,meas}}$ are the corrected and measured $\mathrm{PM_{2.5}}$, respectively, and $\mathrm{f}(T, H)$ is defined in Eq. 2. This physics-based correction was compared to a statistical correction which was a multiple linear correction with terms for relative humidity, air temperature, and dew point temperature. They found that the two correction approaches yielded comparable improvements on $\mathrm{PM_{2.5}}$ readings. Large uncertainties were still present in hourly-averaged readings (mean absolute errors 3-4 $\mu$g m$^{-3}$) but yearly-averaged readings were more accurate (errors less than 1 $\mu$g m$^{-3}$). The Malings et al. (2019) work established that their physics model plus a constant term is reasonably complete, and is the basis for the forward model we will use with our OEM method.

## 2.2 Treatment of the Measurements

This study uses measurements from January to December of 2021 taken by two air quality sensors described below. Daily averages were taken to produce one data point per day. To perform the correction, the measurements were split into four seasons: Spring (March - May), Summer (June - August), Fall (September - November), and Winter (December - February). This binning was chosen to investigate seasonal variance in PM composition on the bulk hygroscopicity.

### 2.2.1 PurpleAir Sensor

Measurements from all public PurpleAir sensors are available for download from the PurpleAir Network map. (https://map. purpleair.com, last access; 27 February 2025). Each sensor has two particle counters, Channel A and Channel B, that report independently for purposes of precision. The data are provided as unfiltered, two-minute readings for each channel. Daily averages of the $\mathrm{PM_{2.5}}$, relative humidity, and temperature measurements were made for our London site on The University of Western Ontario campus (43.01°latitude, -81.27°longitude, 258 m above sea level).

If the raw, 2-minute $\mathrm{PM_{2.5}}$ readings had discrepancies between Channels A and B of more than 4 $\mu$g/m$^3$ or more than 25% of their average, they were removed before performing the daily averages. Any readings that only had a value from one channel were also removed. This procedure eliminated about 3% of the measurements. In addition to this quality control procedure, there were some days, and in one case over three consecutive weeks (August-September 2021), where all measurements were discarded to due to clearly erroneous readings. In some cases the cause of these errors remain unknown, likely due to insects or associated debris in the sensor that had to be removed.

Relative humidity and temperature averages were compared to measurements from the Environment and Climate Change Canada weather station at the London International Airport, about 12 km from the PurpleAir sensor. Due to internal heating and insolation effects, PurpleAir temperature readings can be up to 5.3°C higher and humidity readings up to 24.3% drier than ambient conditions (Barkjohn et al., 2021; Malings et al., 2019). The relative humidity readings from the PurpleAir were consistently about 21% lower than the Airport station values, so a simple correction of adding 21% to each PurpleAir relative humidity measurements was made. The relative humidity measurements before and after the correction was applied are shown in Appendix A. The PurpleAir temperature readings were on average about 2°C higher than the Airport station values, which was not a large enough discrepancy to require correction, since the effect of temperature in the forward model is primarily through the relative humidity (see Section 3.1.3).

### 2.2.2 Ontario MECP Validation Measurements

Measurements from all Ontario Ministry of the Environment, Conservation and Parks (MECP) air quality sites are available on the MECP website (http://www.airqualityontario.com/history/summary.php, last accessed; February 2025). They provide hourly readings of ozone, $PM_{2.5}$, and nitrogen dioxide. For our reference data, we used readings from the London Ambient Air Monitoring Site (42.97°latitude, -81.20°longitude, 244 m above sea level), which is about 6 km away from the PurpleAir sensor. The sensor used at this location is the SHARP 5030, and its $PM_{2.5}$ readings are given as integers values. To test that this rounding was not impacting our correction, we applied various rounding schemes (floor, half round up, and ceiling) to the PurpleAir measurements and redid the corrections. None of these rounding choices had a significant impact on the correction results.

### 2.3 The Optimal Estimation Method

OEM is an inverse method that allows the retrieval of the atmospheric state using a set of measurements and a forward model of the physical system. The forward model, $\mathbf{F}$, is represented as

$$\mathbf{y} = \mathbf{F}(\mathbf{x}, \mathbf{b}) + \epsilon, \tag{6}$$

where $\mathbf{y}$ is the measurement vector, $\mathbf{x}$ is the state vector, the vector which contains the retrieved quantities, $\mathbf{b}$ are additional parameters required by the forward model, and $\epsilon$ is the measurement noise vector.

OEM is based on Bayes' theorem which describes the calculation of conditional probabilities. Bayes' theorem allows the most likely state to be determined consistent with the a priori knowledge, the performed measurement, and their associated uncertainties. The cost function, $\mathbf{C}$:

$$\mathbf{C} = [\mathbf{y} - \mathbf{F}(\hat{\mathbf{x}}, \mathbf{b})]^T \mathbf{S}_y^{-1} [\mathbf{y} - \mathbf{F}(\hat{\mathbf{x}}, \mathbf{b})] + [\hat{\mathbf{x}} - \mathbf{x}_a]^T \mathbf{S}_a^{-1} [\hat{\mathbf{x}} - \mathbf{x}_a], \tag{7}$$

is then minimized to find the optimum value of the retrieval parameters. Here $\mathbf{S}_y$ is the measurement error covariance matrix, $\hat{\mathbf{x}}$ is the normalized state vector, and $\mathbf{x}_a$ and $\mathbf{S}_a$ are the a priori estimate of the state vector and its error covariance matrix. In

**Table 1.** Parameters used in OEM code. The state vector **x** was retrieved by the code while the parameter vector **b** was inputted.

| Variable | Description | Vector |
|---|---|---|
| $c$ | constant linear term | **x** |
| $\kappa_{\text{bulk}}$ | bulk hygroscopicity | **x** |
| $D_p$ | average particle diameter | **b** |
| $PM_{2.5,\text{meas}}$ | $PM_{2.5}$ as measured by PurpleAir | **b** |
| $H$ | relative humidity as measured by PurpleAir | **b** |
| $T$ | temperature as measured by PurpleAir | **b** |
| $\sigma_w$ | surface tension of water | **b** |
| $M_w$ | molecular weight of water | **b** |
| $\rho_w$ | density of water | **b** |
| $R$ | ideal gas constant | **b** |

a successful OEM retrieval, one should be able to slightly vary the a priori estimates without having an effect on the retrieved state. This indicates that the a priori estimates are guiding the solution, not dictating it.

### 2.3.1 Implementing OEM

We implemented OEM using Qpack, a free Matlab function developed for atmospheric instrument simulation and retrieval work (Eriksson et al., 2004) using a forward model similar to that in Eq. 5,

$$\text{PM}_{2.5,\text{corr}} = \frac{\text{PM}_{2.5,\text{meas}}}{\text{f}(T,H)} + c. \tag{8}$$

Our forward model does not include the constant $a$ appearing in the form of the model described by Malings et al. (2019). The constant factor is not required in OEM as we are directly retrieving the hygroscopicity of the bulk aerosol. Descriptions of all variables, along with their model parameter type, are given in Table 1.

We took the measurement vector **y** to be the reference $PM_{2.5}$ readings from the MECP site, since the goal was to have the corrected data match these reference readings. The a priori state vector consists of a constant linear term and a bulk hygroscopicity. The a priori bulk hygroscopicity was informed by Cerully et al. (2015). The a priori constant linear term was estimated from simple straight line fits to the measurements. The components of the a priori state vector and their errors are presented in Table 2.

**Table 2.** The components of the a priori state vector and their errors used in OEM code. The errors were used to create the a priori covariance matrix.

| Parameter | Value |
|---|---|
| $c$ ($\mu$g m$^{-3}$) | $1.9 \pm 0.66$ |
| $\kappa_{\text{bulk}}$ | $0.25 \pm 0.025$ |

## 2.4 Statistical Metrics used to Assess the correction

The accuracy of the correction was assessed using Mean Absolute Error (MAE) and bias. MAE is used to assess how well, on average, a data set agreed with the reference data. It is calculated as

$$\text{MAE} = \frac{1}{n} \sum_{i=1}^{n} |r_i - \hat{r}_i| \tag{9}$$

for $n$ measurements of PurpleAir readings ($r$) and reference readings ($\hat{r}$). Similarly, the bias of each data set is used to assess the systematic differences between data sets and was calculated as

$$\text{bias} = \frac{1}{n} \sum_{i=1}^{n} (r_i - \hat{r}_i). \tag{10}$$

Lower values of MAE and bias indicate better agreement between our data and the reference data. We also used the adjusted coefficient of determination, adjusted-R$^2$, to assess how well our corrected data correlated with the reference data.

## 3 Evaluation of the OEM Model

### 3.1 Sensitivity Analysis

One of the many advantages of OEM is the ability to investigate the sensitivity of the retrieval to each model parameter. For each model parameter, the error covariance matrix, $\mathbf{E}$, is given by

$$\mathbf{E} = \mathbf{G} \cdot \mathbf{J}_b \cdot \mathbf{S} \cdot \mathbf{J}_b^{\mathbf{T}} \cdot \mathbf{G}^{\mathbf{T}}, \tag{11}$$

where $\mathbf{G}$ is the gain matrix,

$$\mathbf{G} = \frac{\partial \hat{\mathbf{x}}}{\partial \mathbf{y}}, \tag{12}$$

$\mathbf{J}_b$ is the Jacobian for the parameter represented by $b$,

$$\mathbf{J}_b = \frac{\partial \mathbf{F}}{\partial b}, \tag{13}$$

and $\mathbf{S}$ is the uncertainty covariance matrix (Rodgers, 2000). We will use these equations to investigate the impact of particle diameter, temperature, and relative humidity on the retrieval.

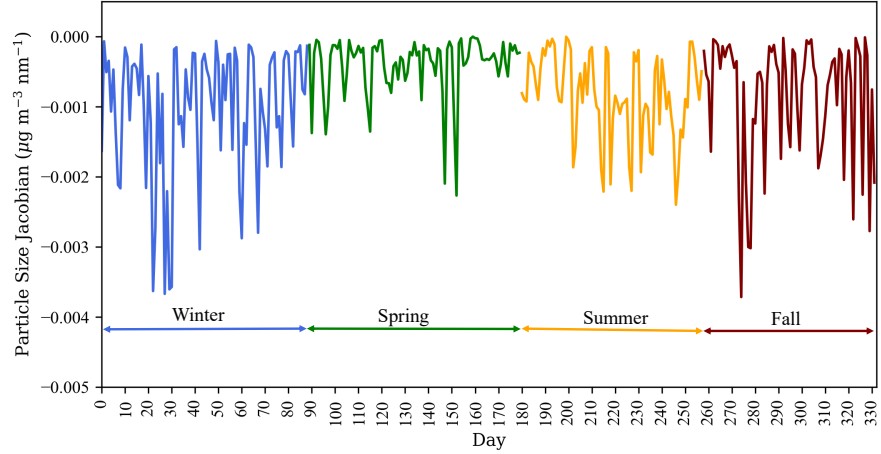

**Figure 1.** A time series of the sensitivity of the forward model to the particulate diameter parameter. The y-axis depicts the change in the corrected PM$_{2.5}$ value for every 1 nm change in the assumed particle diameter.

### 3.1.1 Sensitivity to Average Particle Diameter

We tested the sensitivity of the model to the average particle diameter by investigating the Jacobian given in Eq. 13, where the parameter, *b*, is the particle diameter. This Jacobian shows that the change in the corrected PM$_{2.5}$ for a change in particle size of 1 nm is three orders of magnitude smaller than the PM$_{2.5}$ measurements, and thus insignificant (Fig. 1). Due to the low sensitivity of the forward model to particle diameter, the choice of $D_p$ had a negligible effect on the total retrieval error. Hence, even large uncertainties in average particle diameter assumed do not impact the retrieval.

This result is significant because it indicates that additional instrumentation that measures $D_p$ is not needed in order to apply the physical model. Using a general estimate of average particle diameter, in our case a value of $D_p = 200$ nm, can be recommended for future applications of this model.

### 3.1.2 Sensitivity to Temperature

We investigated the behaviour of the physical model in response to temperature to check if the PurpleAir sensors temperature measurements are of sufficient quality for the intended use. The Jacobian is shown in Fig. 2. We found that the forward model is not sensitive to temperature and changes in temperature throughout a range of typical London annual temperatures (-20 °C to 35 °C), would not impact the retrieval significantly. Therefore, the PurpleAir temperature measurements may be used without correction. It should be noted that the physical model is still sensitive to temperature through the relative humidity dependence, discussed in the next section.

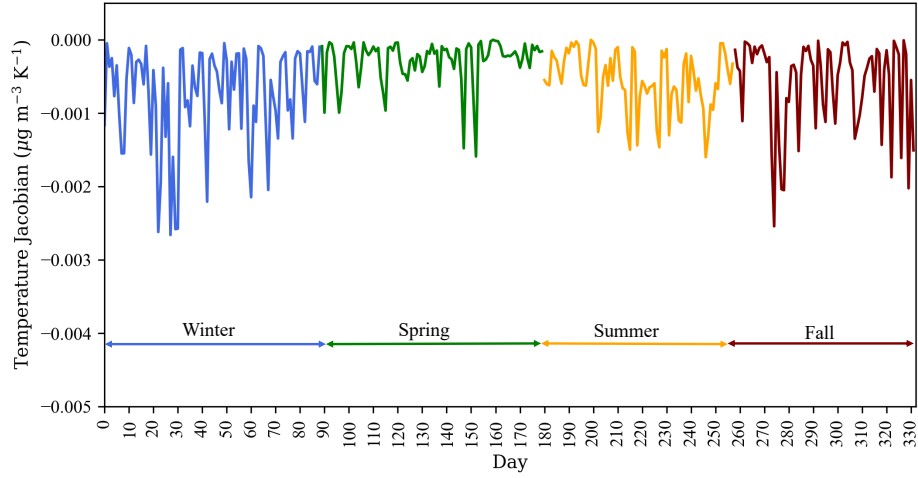

**Figure 2.** A time series of the sensitivity of the forward model to the temperature parameter. The y-axis depicts the change in the corrected $PM_{2.5}$ value for every 1 K change in the temperature.

### 3.1.3 Sensitivity to Relative Humidity

We also investigated the sensitivity of the forward model to relative humidity. The Jacobian for relative humidity is shown
in Figure 3a. The Jacobian is large enough to indicate that the model is sensitive to relative humidity. The Jacobian varies
day-to-day due to its correlation with the $PM_{2.5}$. Fig 3b shows this relation.

Since the model is sensitive to relative humidity, we recommend correcting the PurpleAir relative humidity measurements
before use. For this dataset, a constant correction of 21% was sufficient to make the PurpleAir measurements better agree with
measurements from the nearby Aiport meteorological station, but this correction may not be applicable for other sites or time
periods. The relative humidity measurements before and after the correction compared to the Airport measurements are given
in Appendix A.

### 3.2 OEM Correction Results

After using OEM to retrieve the hygroscopic growth factor and the constant term, we applied the physical model (Eq. 8) to
correct the PurpleAir measurements from each season. The physically-corrected and raw $PM_{2.5}$ measurements are shown in
Fig. 4. For comparison, the PurpleAir measurements were also corrected using multiple linear regression (MLR) as was done
for example by Barkjohn et al. (2021). The MLR result is also shown in Fig. 4. Details of the MLR correction are given
in Appendix B. The physical correction, shown in red, succeeded in bringing the raw measurements closer to the reference
measurements and performs similarly to the statistical correction shown in blue. The cost function of the OEM retrieval is
around 5.5, signifying a good fit without overfitting, which occurs when the cost function is less than 1.

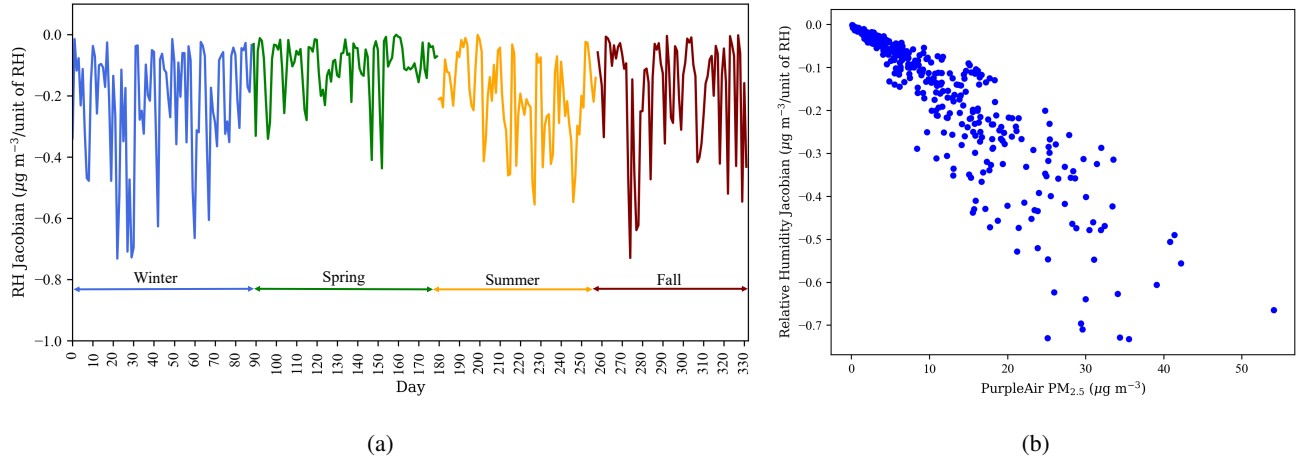

**Figure 3.** (a) A time series of the sensitivity of the forward model to the relative humidity (RH). The y-axis depicts the change in the corrected PM$_{2.5}$ value for every 1% change in RH. (b) The correlation between the RH Jacobian and the PM$_{2.5}$ measured by the PurpleAir, with each point representing one day.

The MAE and $R^2$ values from both models are given in Table 3. The physical correction succeeded in reducing the MAE and bias of the raw data. Averaged over all seasons, the MAE was reduced by 69% and the magnitude of the bias was reduced by 95%. Overall, the spring data was the best of the physically-corrected results for all of our metrics (MAE, bias, and $R^2$). Although both models performed similarly, the statistically-corrected data had smaller MAE and slightly higher values of $R^2$. It can be concluded that the statistical correction had better overall performance, while the physical correction allows a new
physical data product to be retrieved with slightly poorer PM$_{2.5}$ correction.

**Table 3.** MAE of all raw, physically- corrected, and statistically- corrected measurements along with the R$^2$ value of the physical and statistical corrections

| Season | Raw MAE $\mu$g m$^{-3}$ | Raw R$^2$ | Physical MAE $\mu$g m$^{-3}$ | Physical R$^2$ | Statistical MAE $\mu$g m$^{-3}$ | Statistical R$^2$ |
|--------|------|------|------|------|------|------|
| **Spring** | 3.48 | 0.61 | 1.38 | 0.78 | 1.05 | 0.87 |
| **Summer** | 6.95 | 0.42 | 1.98 | 0.72 | 1.74 | 0.78 |
| **Fall** | 5.00 | 0.40 | 1.59 | 0.61 | 1.50 | 0.66 |
| **Winter** | 6.90 | 0.46 | 1.78 | 0.75 | 1.34 | 0.75 |

To investigate the impact of relative humidity on the correction, the bias of the physically-corrected data from each season are plotted as a function of relative humidity (Fig. 5). This was done to illustrate how the correction behaves at higher humidities. We can see that the physical correction tends to over correct the data when the relative humidity is above about 65% as indicated

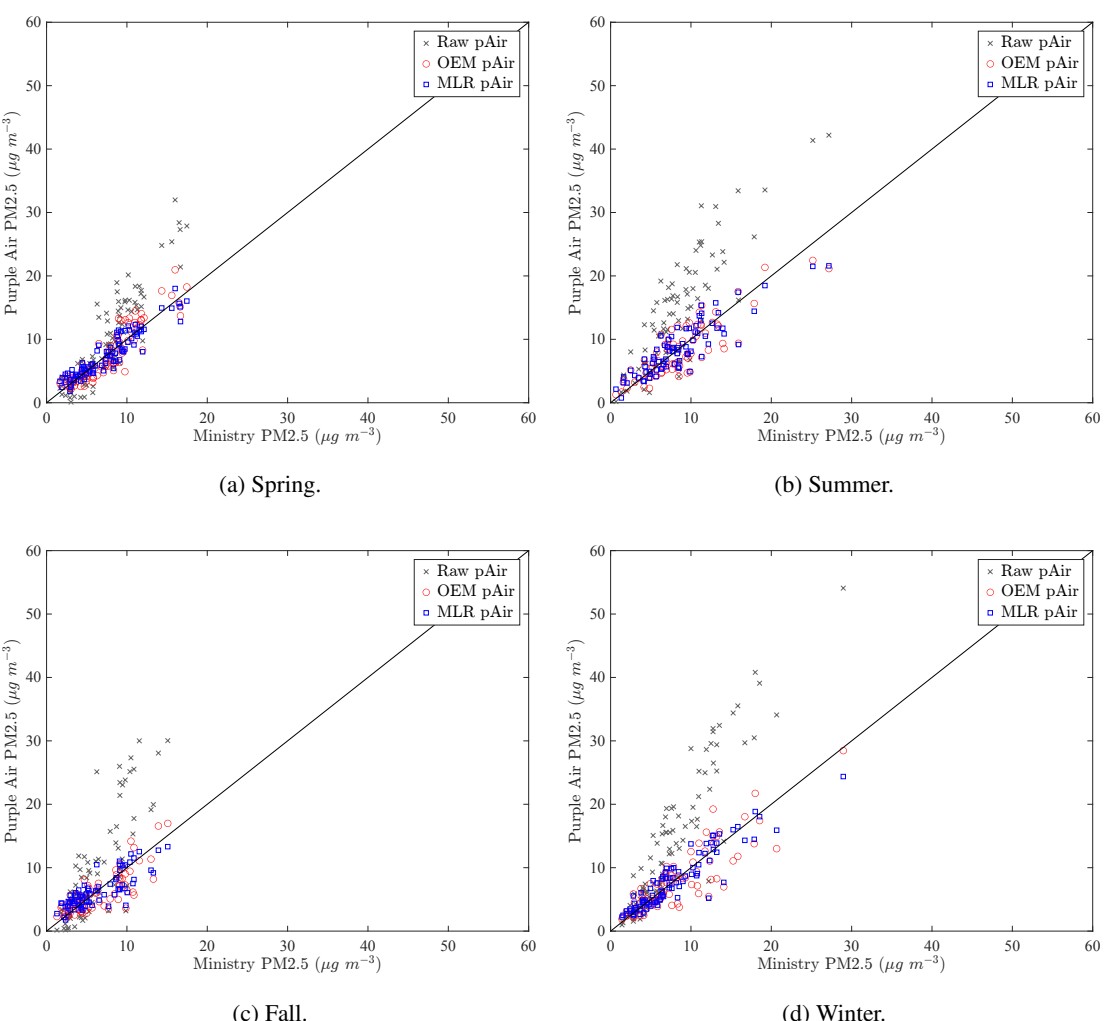

**Figure 4.** Daily-averaged $PM_{2.5}$ as measured by MECP vs PurpleAir. Data correction using the Optimal Estimation Method (OEM) in red, raw data in grey, statistically corrected data using a multiple linear regression (MLR) in blue, and 1:1 line in black. The horizontal axes are $PM_{2.5}$ readings by the MECP reference sensor, and the vertical axes are averaged $PM_{2.5}$ readings from the PurpleAir sensor. The location of each point in the plots signifies one day of $PM_{2.5}$ measurements by the MECP sensor and the PurpleAir sensor measurement for the same day averaged from two-minute readings.

by a negative bias. This effect is not observed consistently with the statistically corrected data. Thus, corrections using OEM at higher values of relative humidity may be insufficient. Mathieu-Campbell et al. (2024) suggests a clustering approach is more effective at correcting Purple Air measurements in high humidity conditions, which allows the non-linearity associated with hygroscopic growth to be captured. The average daily bias is less than $1\ \mu\text{g}\,\text{m}^{-3}$ for both correction models.

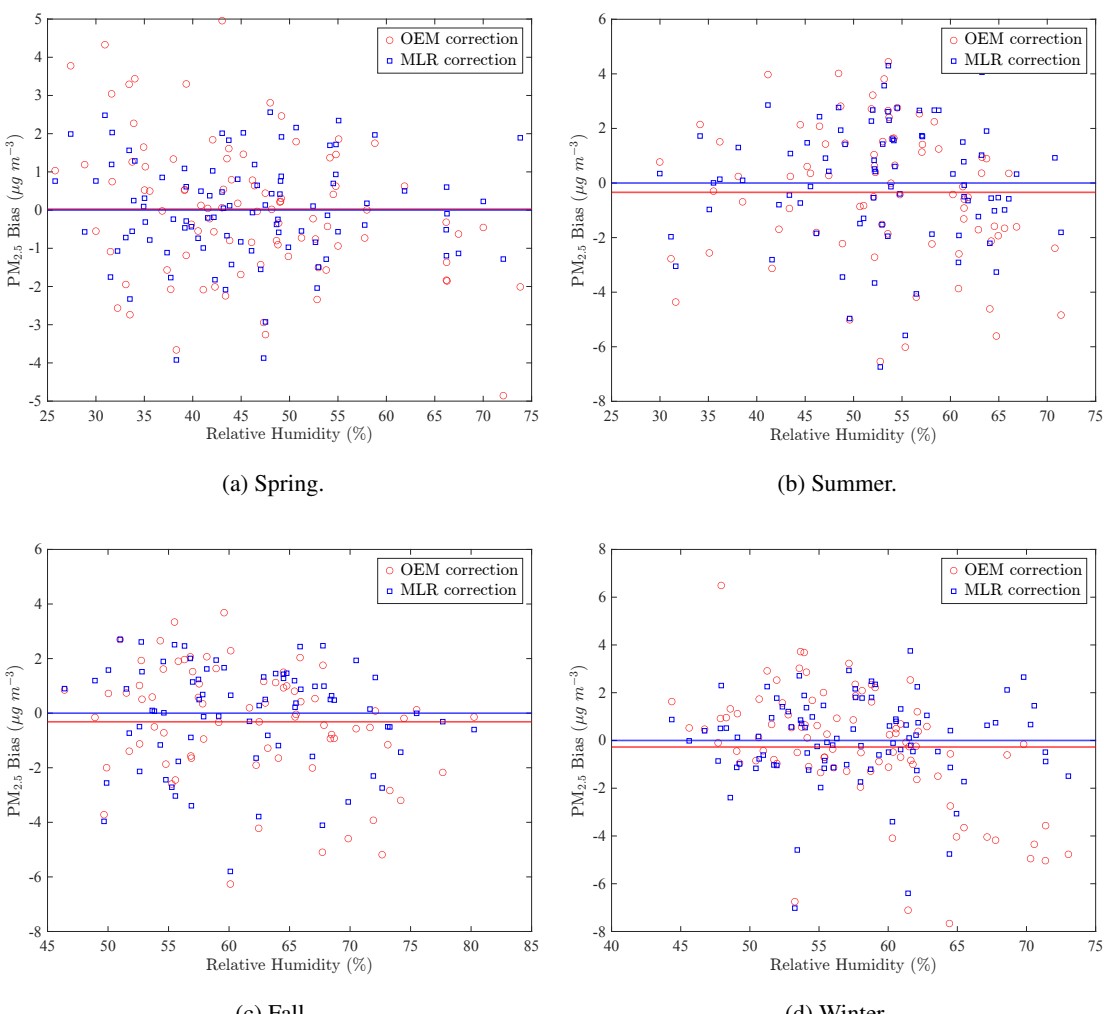

**Figure 5.** The bias in the PM$_{2.5}$ measurements corrected using the Optimal Estimation Method (OEM) shown in red, and using a multiple linear regression (MLR) shown in blue, as a function of relative humidity. The red horizontal line shows the average bias from the OEM correction, and the blue horizontal line is the average bias from the MLR correction.

### 3.3 Bulk Hygroscopicity Results

The bulk hygroscopicity of particulates is one of the parameters of the physical model retrieved through OEM. The seasonal values of bulk hygroscopicity and the associated hygroscopic growth factor retrieved are shown in Fig. 6a and 6b respectively. The error on bulk hygroscopicity is the sum of the observation error from OEM and the error due to relative humidity calculated using Eq. 11, assuming that the relative humidity is known within 2.5% uncertainty. This is a reasonable uncertainty for a relative humidity sensor to achieve using measurements from a co-located, calibrated weather station. For our pilot study, the relative humidity values were not of sufficiently high quality, as previously discussed.

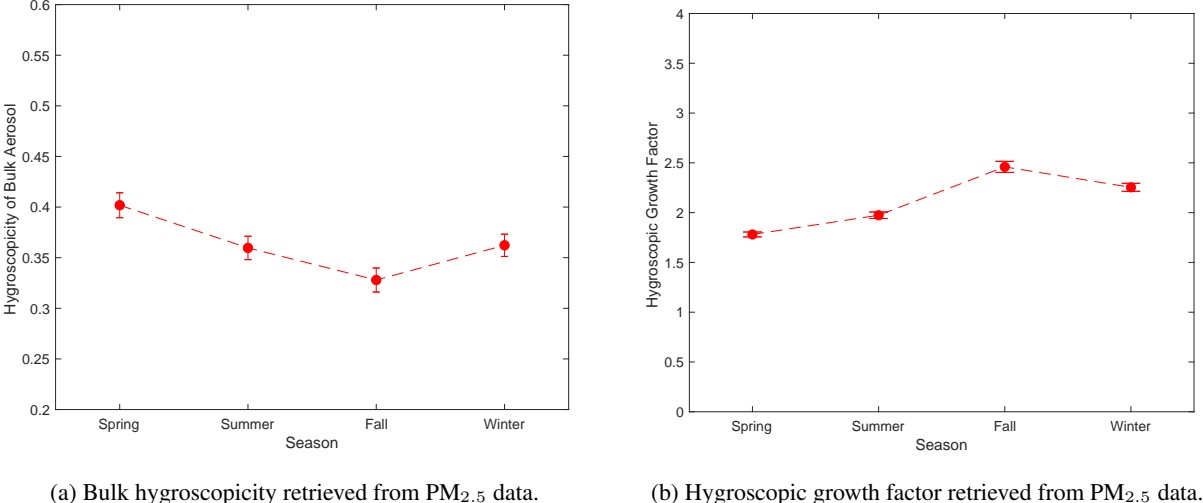

(a) Bulk hygroscopicity retrieved from PM$_{2.5}$ data.      (b) Hygroscopic growth factor retrieved from PM$_{2.5}$ data.

**Figure 6.** Bulk hygroscopicity and the hygroscopic growth factor retrieved through OEM for each season. Uncertainties are based on the assumption that relative humidity is known within 2.5% uncertainty.

### 3.3.1 Investigating Retrieved Bulk Hygroscopicity

The retrieved values of hygroscopicity of bulk aerosol are between 0.33 and 0.4 as shown in Fig. 6, which are consistent with values in the literature (Cerully et al., 2015; Petters and Kreidenweis, 2007). The hygroscopicity of bulk aerosol is the smallest in the Fall and largest in the Spring. This is consistent with the result from (Akpootu and Gana, 2013) that describes an inverse relationship between hygroscopicity and relative humidity. Our results show highest relative humidity in the Fall and the lowest in the Spring, with the hygroscopic growth factor varying proportionally as expected. The error due to relative humidity is on average 4% of the total error, which means that it is small in comparison to the observation error. Therefore, even though it was found in Section 3.1.3 that the forward model (the corrected PM$_{2.5}$) is sensitive to relative humidity, our retrieved value for bulk hygroscopicity is not very sensitive to relative humidity. The results shows that our method may have the potential to estimate bulk hygroscopicity.

## 4 Limitations of Our Approach

One limitation of our method was that the reference instrument was not co-located with our PurpleAir sensor. From observations of the spatial spread of PM$_{2.5}$ from the PurpleAir website, we noticed that regions in close proximity to the London site used for this study follow the same trends in PM$_{2.5}$ and generally have very similar readings. It is due to this that we were comfortable carrying out this correction with our reference site approximately 6 km away. We also attempted to work around this limitation through daily averaging, which should allow two sites in the same city to reach similar values of PM$_{2.5}$, but it is still impossible to know exactly the effect that this limitation could have had on our study.

Another limitation is regarding the correction of daily-averaged data sets. The main purpose of this correction is to correct for effects of relative humidity. These effects cannot be fully represented when taking daily averages since relative humidity varies significantly throughout the span of 24 hours and averaging greatly smooths these variations. Therefore it is undetermined if the daily-averaged data sets fully encapsulate the model of hygroscopic growth in the presence of humidity. Also, This study did not include any extreme events, such as wildfires, which would increase the $PM_{2.5}$ significantly, so it is not known how our method would be affected by unusually high concentrations.

Finally, this method requires high quality measurements of relative humidity, beyond what the Purple Air sensor is capable of. Best conditions for the application of this technique would include a co-located, calibrated weather station and close proximity to the correction source.

## 5   Conclusions

We applied a physical model based on the hygroscopic growth of particulates to correct $PM_{2.5}$ measurements from a PurpleAir sensor and showed that it is possible to estimate the hygroscopic growth as part of the sensor calibration using the Optimal Estimation Method. We corrected daily-averaged data for one year split into four seasons. The physical correction reduced average daily MAE and bias from $5.58\,\mu g\,m^{-3}$ to $1.68\,\mu g\,m^{-3}$ and from $4.75\,\mu g\,m^{-3}$ to $-0.23\,\mu g\,m^{-3}$, respectively. The physical model tended to over correct data points with daily-averaged relative humidity above approximately 65%. This relative humidity bias was not seen in the statistical correction, which reduced the average bias to $0\,\mu g\,m^{-3}$ (due to the statistical nature of linear regression) and average MAE to $1.46\,\mu g\,m^{-3}$.

The physical correction did not perform quite as well as statistical correction, but it did provide insight into the physical model of hygroscopic growth of particulates. We found that the average particle diameter does not need to be measured and can simply be estimated for future implementations of this model. This makes the physical model applicable to more sites that do not have access to these measurements. Additionally, we were able to use OEM to retrieve reasonable values of bulk hygroscopicity ranging from 0.33 to 0.4. Furthermore, our method is extremely fast computationally, making it ideal to apply to "real time" situations such as air quality maps like the hourly $PM_{2.5}$ UNBC/ECCC map by Nilson and Jackson (https://aqmap.ca/aqmap, last access; 27 February 2025).

The main limitation of this study was our inability to access co-located reference measurements. We encourage researchers with dedicated air quality observatories with more sophisticated, co-located equipment to test our method and compare our bulk hygroscopicity estimates with other techniques. Furthermore, it would be of interest to investigate the inability of the physical model to represent $PM_{2.5}$ at high levels of relative humidity. Finally, it should be noted that the OEM method could in principle retrieve the individual values of the hygroscopicity as well as the bulk values as in the current retrieval.

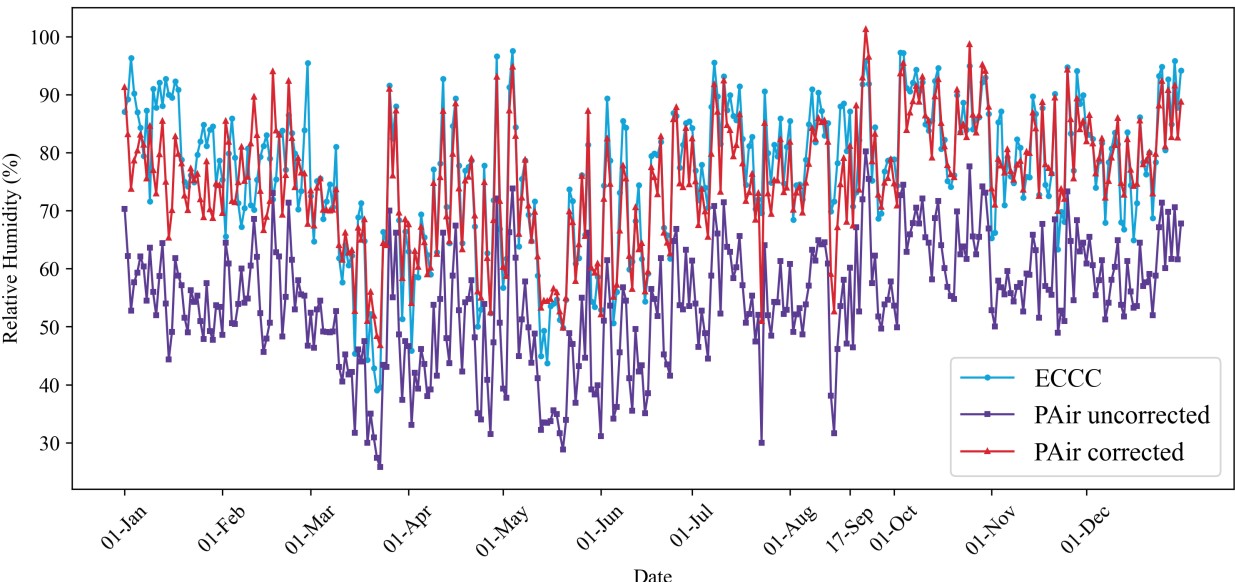

**Figure A1.** A time series of the relative humidity measured by the Purple Air sensor before and after correction, compared to the ECCC reference measurement taken at the London Airport.

## Appendix A:  Humidity Correction

PurpleAir sensors are known to have high temperature readings and low relative humidity readings due to internal heating (PurpleAir, 2021). A simple correction factor of adding 21% to all PurpleAir measurements sufficed to correct the PurpleAir relative humidity for our purposes. The raw and corrected relative humidity readings across the year of data used in this study
are shown in Fig. A1 along with readings from the official Environment and Climate Change Canada (ECCC) weather station.

## Appendix B:  Statistical Correction

The forward model equation for the statistical calibration was as follows:

$$\mathrm{PM}_{2.5,\mathrm{corr}} = a\mathrm{PM}_{2.5,\mathrm{meas}} + b\mathrm{H} + c, \tag{B1}$$

where $a$, $b$, and $c$ are constants. Their values for each data set are shown in Table B1.

**Table B1.** Statistical model coefficients and errors.

| Season | $a$ | $b$ $\mu$g/m$^3$ | $c$ $\mu$g/m$^3$ |
|--------|-----|------------------|------------------|
| Spring | 0.46 ± 0.02 | -0.07 ± 1.34 | 6.25 ± 0.95 |
| Summer | 0.49 ± 0.03 | -0.10 ± 2.77 | 6.65 ± 1.96 |
| Fall | 0.33 ± 0.03 | -0.06 ± 2.96 | 6.85 ± 2.40 |
| Winter | 0.42 ± 0.02 | -0.014 ± 3.16 | 2.68 ± 2.44 |

*Code availability.* The OEM retrieval used for this study is part of the ARTS package and can be downloaded at https://github.com/atmtools.

*Data availability.* The data set used is available on the Zenodo database at https://doi.org/10.5281/zenodo.14146969.

*Author contributions.* JP was responsible for the initial assembly and processing of the Purple Air and Ministry measurements and writing an initial report of the work. ET performed much of the reprocessing of the data as well as contributions to the manuscript. RJS supervised both students, helped in the implementation and coding of the OEM method, and wrote an initial draft of the manuscript.

*Competing interests.* The authors declare that they have no conflict of interests.

*Acknowledgements.* We thank the Reviewers for their many insightful comments which improved our explanations and presentation of the results.

This project has been funded in part by the National Science and Engineering Research Council of Canada through a Discovery Grant (Robert J. Sica) and by a Canadian Space Agency FAST grant (Kimberly Strong, PI).

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
