# Peer review of "Retrieval of Bulk Hygroscopicity From PurpleAir PM2.5 Sensor Measurements"

_EGUsphere, 2024_

## Referee Comment (RC2)

**Specific Points**

1. **Abstract**
   1.1. Line 3: remove "(empirical)" in favour of just "statistical" or "empirical"
   1.2. Line 10: clarify the bias metric used similar to how it was done for MAE
   1.3. Line 12: "using our OEM retrieved allowed" - replace retrieved with retrieval?
   1.4. Line 13: include the accepted ranges in parenthesis here if possible

2. **Introduction**
   2.1. Line 21: add proper citation for map.purpleair.com and move to references
   2.2. Line 22 and 24: It is incorrect to say the PurpleAir monitor makes particulate "measurements" - they estimate the concentrations based on measurements of light scattering and an assumed particle composition
   2.3. Line 25: the term "Low-cost sensors" is used but not defined for the reader
   2.4. Line 35: Barkjohn et al. expanded on this work in this publication: https://www.mdpi.com/1424-8220/22/24/9669
   2.5. Line 44: Suggest including https://amt.copernicus.org/articles/15/3315/2022/ as a reference as it compares many of the cited models with sites across Canada/USA
   2.6. Line 45: remove "(also called empirical)" in favour of just "statistical" or "empirical"
   2.7. Suggest adding paragraph(s) describing the OEM method and hygroscopic growth

3. **Methodology**
   3.1. What time period is the data from? I believe 2021 based on the comment in this section on data removed for a period in August 2021, but it should be clearly stated what date range was used.
      3.1.1. Suggest adding a study site paragraph to start this section.
   3.2. Section 2.1 belongs mostly in the introduction as it is a review of what was done in another study. Lines 75 and 76 should be expanded on here instead with specifics on what was done in this study
   3.3. Line 74: what total is "or about 30%" in reference to?
   3.4. Line 79: citation for PurpleAir is missing
   3.5. Line 80: having two sensors is for precision, not accuracy
   3.6. Line 84: clarification is needed whether the A/B comparison was done before or after averaging to daily averages
   3.7. Line 86: "typically eliminated about 3% of measurements" - unclear what this 3% applies to, each day?
   3.8. Line 90: "due to internal heating and insolation effects"
   3.9. Line 93: In my experience, the bias in PA RH has a diurnal cycle due to insolation effects on the temperature within the unit. Given this, adding 21% to PA RH may be okay for this dataset but it may not be transferable between sites/time/averaging periods. This should be emphasised in the discussion/conclusion, and I would suggest including justification for the 21% adjustment in the results (ex. a scatter of RH from both monitors before and after)
   3.10. Line 94: "were about 2C high" is vague
   3.11. Line 101: link should be cited properly and moved to the reference list
   3.12. Lines 108 - 121: this paragraph belongs in the introduction

3.13. Line 108: suggest improving the paragraph transition here. This paragraph should start with the *measurement/monitor* differences, not the *price* differences.

3.14. Line 111: the plantower sensors independently produce particle counts and concentration estimates using two separate proprietary algorithms (ie. concentrations are not derived from the particle counts). See: https://amt.copernicus.org/articles/13/6343/2020/

3.15. Line 117: I don't believe "swelling" is the correct term. The water accretes on the surface of the particle; swelling implies the absorption of water by the particle

3.16. Line 119: "detect higher concentrations of larger-diameter particles" is incorrect - due to the hygroscopic growth of the particles the sensors detect higher scattering and estimate a higher concentration. The assumed particle density does not change.

3.17. Lines 123 - 135: belongs in the introduction

3.18. Line 156: what bias metric was used?

**4. Results**

4.1. Line 163: Remove "One model parameter is the particle diameter."

4.2. Line 171: define "reasonable ambient range"

4.3. Figure 1:

    4.3.1. suggest increasing font size as it is difficult to read even with zooming

    4.3.2. Make the x/y axes have the same limits so seasons can be compared visually. As it is now, the Raw fall values look biased higher than that for the winter panel, however the winter axes go out to 60 ug/m3 whereas the fall axes go out to 40 ug/m3

    4.3.3. "OEM" and "MLR" need to be defined in the figure caption

4.4. Line 175: what form was the MLR? And what were your coefficients? This needs to be discussed in the methods and results

4.5. Lines 177-187: use these sections to describe the results displayed in the figures, not to describe the presentation of the figure. For example, "The raw PurpleAir observations tended to be biased higher than the Ministry PM2.5, which worsens as concentrations increase."

4.6. Line 188: "The physical calibration has a tendency to over-correct at high relative humidity" - is this true? Figure 2 has high and low RH on both sides of the 1:1 line for all seasons. I don't think this is sufficient justification to disregard higher humidity values.

4.7. Figure 2:

    4.7.1. Increase font size as it is difficult to read without zooming

    4.7.2. The figure caption should state that these are daily observations

    4.7.3. The colour scale used makes it difficult to see the mid-range values, this makes the extremes visually stand out. I would suggest binning the RH values into low (<55%) moderate (55- 65) and high (> 65%) and using 3 easily differentiable colours. This would also make it more clear what "higher values of relative humidity" (Line 190) entails.

4.8. Line 195: the PurpleAir spec sheet lists an accuracy tolerance of 3% (https://www2.purpleair.com/products/purpleair-pa-ii)

4.9. Figure 3:

    4.9.1. Replace "PM2.4" with "PM2.5" for both (a) and (b)

      4.9.2. The error bars are not clearly visible on (b) - this should be mentioned in the results section

      4.9.3. (b) has "Month" as an x axis label not "Season"

  4.10. Additional recommendations:

      4.10.1. A daily mean time series of the observation data would benefit this paper, especially for justifying the bias-adjustment of 21% for the PurpleAir RH.

      4.10.2. Instead of Figure 2, try RH on the x axis and daily mean bias on the y. That could more clearly show if high RH has an effect on the PurpleAir bias.

**5. Discussion**

  5.1. Lines 199-201: this is an important finding that could be highlighted in the abstract

  5.2. Lines 209 - 211: something like this would be great for the results section

  5.3. Line 217: particle composition/age also varies seasonally as well and is an important factor. The optical properties and hygroscopicity of particulates from a residential wood stove that have not been airborne for long will differ from those from wildfire smoke that travelled from western Canada to eastern Canada. Another factor could be the concentration ranges within each season - concentrations of PM2.5 tend to be lower in Spring/Fall due to less periods of stagnation

  5.4. Line 218: "biggest" is vague and it is unclear to me what supports this claim

  5.5. Line 220: "are more strongly affected" is vague

  5.6. Line 228: it is not clear to me how the apparent seasonal differences in overcorrection at high humidities indicates that the physical model does not perform well at high humidities.

  5.7. Line 231: Table 2 and the summary of it belongs in the results, not discussion

  5.8. Lines 232-233: "The statistically-calibrated data consistently had no bias" - this seems suspect to me (potentially a result of not splitting training/testing data), and is vague. Where was this presented in the results?

  5.9. Table 2: What about the raw R squared? Did the physical/statistical models improve/worsen the correlation?

  5.10. Line 236: "greatest" is subjective and was not statistically evaluated - recommend removing this sentence

  5.11. Line 237: "it is known" should be replaced with "we noticed" and the sentence should be clarified that it is for this site/region - PM2.5 can have dramatic spatial variation

  5.12. Additional limitations

      5.12.1. It is a concern that only one year for a single PurpleAir/Ministry pair was used- are these results transferable to other areas and concentration ranges?

      5.12.2. The data were not split into training and testing datasets, likely overfitting the model and producing overoptimistic performance measurements.

      5.12.3. The concentration range is fairly moderate, it would be interesting to see the performance during wildfire smoke events where daily mean PM2.5 can exceed 100 ug/m3 (more than twice of what was observed at this site/period)

**6. Conclusions**

  6.1. Line 258: include the reasonable ranges in parenthesis here if possible

  6.2. Line 260: use the updated url for aqmap (https://aqmap.ca/aqmap), and move to references with a proper citation

---

## Author Comment (AC1)

Response to Reviewer 1
Retrieval of Bulk Hygroscopic from PurpleAir Sensors, Psotka et al., (egusphere–2024–3618)

**Major Comments**

**Reviewer:**
This paper presents an interesting approach for extracting aerosol properties from low-cost air quality sensor data. While the method is novel and the results are promising, I would suggest a few additional analyses and clarifications in the work.

First, and most significantly, it seems that the same data are used for both calibrating the OEM method (i.e., extracting the aerosol properties) and for evaluating the performance of this method. The same holds true for the MLR method against which the OEM is being compared. I don't think that this is a fair way to do these comparisons. If possible, I would suggest separating the data from each season into distinct sets for calibration and evaluation. This separation could be done randomly or (my preference) using e.g. one month of data to run the OEM (and to calibrate the MLR) and the other months to evaluate the calibration performance. I prefer this latter approach since it is more realistic to how these sensors might be used in practice, i.e., calibrated near a reference for some period of time and then deployed to another site. Overall, I think this will be a more realistic evaluation of the method and its strengths/limitations compared to the "traditional" MLR.

**Response**:
We apologize for the confusion we caused with both reviewers. We should not have said we were deriving a calibration for the measurements. We did not do that. Rather, we are correcting local measurements from a single site to demonstrate the potential for our method to be applied to correct single-day measurements (e.g. on the University of Northern British Columbia Air Quality map (https://cyclone.unbc.ca/aqmap/v2/)), or in the future used to calibrate networks of PurpleAir sensors as have been done in works like Mailings et al. 2019 and  Barkjohn et al. 2021. Then, what we propose above (e.g. a training set) would be the appropriate way to develop a calibration.

The procedure of using a training set for a retrieval is not typically done when solving inverse problems, as the forward model is a complete (as possible) description of the geophysical situation and the instrument function. What is problematic in OEM is how good is the model fit, that is, the value of the cost function. If the cost function is less than 1, then the model is overfitting the measurements (bad!). If the cost function is too high, say (arbitrarily) 10, the model is missing something. A perfect fit of the model to the data would be a value of 1 for the cost function. In our case, the cost values are around 5, meaning we are doing a good job of describing the measurements without overfitting them.

**Reviewer:**
Second, there are some comments that the method is quite sensitive to the relative humidity, and that the PurpleAir relative humidity data are of insufficient quality. Can more be said about this? I would suggest, for example, testing the method with the relative humidity as measured by PurpleAir, but using a higher relative uncertainty for these measurements in the covariance matrices within OEM. Is the OEM method still able to resolve the aerosol properties in such cases? While accurate humidity data from a nearby whether station may be available, it would be useful to also see how well the calibration can perform using only the information from the PurpleAir itself (while understanding the relatively lower quality of these data).

**Response**: The Jacobian of the forward model with respect to relative humidity is now presented in Figure 1 to quantitatively show that the PM$_{2.5}$ correction given by the forward model is sensitive to relative humidity. We also included more discussion in section 3.5.1 on the retrieved bulk hygroscopicity error, which we found to be less sensitive to relative humidity. A benefit of using an optimal estimation method is that the sensitivity of the retrieved parameters (here hygroscopicity) to a model parameter (here temperature, particle diameter, relative humidity) can be explicitly calculated.

**Reviewer**:
Third, the OEM method requires appropriate prior terms, and especially appropriate uncertainties (parameterized in covariance matrices) for these terms. While there is some discussion of the sensitivity of the method to various parameters, there is little quantitative information here which would help other researchers understand the applicability of these findings to their work. I would suggest describing how the measurement error covariance matrix and the state vector covariance matrix were defined in more detail. Furthermore, I would suggest presenting the results of any sensitivity analyses conducted for these terms, possibly in a supplement or appendix.

**Response**: The a priori state vector and its error are now given in Table 2. The Jacobian for each of the parameters (temperature, particle diameter, and relative humidity) are now shown in Figures 1-3. Section 3.1 describes the sensitivity analysis that we performed.

**Other Comments:**
I also have several specific comments and suggested corrections, listed here:

Line 20: Note that these data are still publicly available, but not freely available.

**Response**: The data is freely available on the PurpleAir website as shown here:
https://api.purpleair.com/#api-sensors-get-sensors-data.

Line 25: Please provide a source or reference for the description of the operating principles.

**Response**: Ardon-Dryer et al. (2020) and Ouimette et al. (2022) were added as references describing the Plantower sensors used by the PurpleAir device.

Line 30: "was" should be "were".

**Response**: Done.

Line 43: The calibration is not only applicable to "their" sensor, but also any other sensor, within the range of conditions under which the calibration was created and validated.

**Response**: Done.

Lines 71-72: The meaning of this sentence is unclear. I think you might intend to say that the physics-based correction model which was just described was also compared with a purely statistical correction model incorporating multiple linear regression terms. Please consider rephrasing if this is indeed what you meant.

**Response**: Done.

Line 87: What is meant by "clearly erroneous readings"? Is this the result of the same quality control procedure just described, or are these data erroneous for different reasons?

**Response**: This is separate from the quality control procedure described. This was clarified in the text in the second paragraph of Section 2.2.1. This was a one-time event likely due to the sensor being jammed with insects.

Section 2.2.3: Not a necessary change, but I would suggest moving most of this information to the introduction, as it is background detail and motivation for the study. The specific model of reference instrument used can be mentioned in Section 2.2.2 instead.

**Response**: Done.

Section 2.3: Values for the a-priori state vector and the error covariance matrices should be provided; this can be done in a supplement or appendix. Some argument should be provided about how these values were selected as well, so that others can make appropriate decisions when replicating this.

**Response**: The a priori are now listed in Table two and the sources for these values are referenced in Section 2.3.1.

Section 3.1: Are there any quantitative results which can be presented based on this analysis to support the conclusions that particle diameter and temperature are of lesser importance? For example, relative magnitudes of terms in the error covariance matrix?

**Response**: As described previously, the Jacobians have been added to section 3.1.

Line 176: remove "from".

**Response**: Done.

Section 3.2.1: Are the same results observed for the empirical calibration, or is this over-correction unique to the physical calibration? There is a comment addressing this in lines 229-230; maybe this could be moved up and expanded on. The comment seems to suggest that, based on the data, the calibration dependence on humidity should in fact be linear, as opposed to nonlinear as in the physical calibration. This could potentially mean that humidity is impacting the sensor performance in more complex ways than just the hygroscopicity of the particles.

**Response**: This over-correction is more prominent in our physical calibration, but it has been observed in other studies with statistical corrections as well. Figure 5 was added to make this observation more clear. We also added a reference to a recently published paper, Mathieu-Campbell et al. (2024), describing a new way to correct high humidity measurements in Section 3.3.

Line 196: I would suggest using the measured humidity from the PurpleAir as your input and calibrating the uncertainty for this term using your comparison to a nearby weather station. This could give a sense of how robust the measurements are to the data quality of the internal humidity sensor, which you have noted is not high.

**Response**: We now present the Jacobian for relative humidity and expand upon how bulk hygroscopicity is affected by the error in relative humidity as previously described.

Lines 218-219: This argument seems to contradict the previous statements that the method is not sensitive to particle diameter.

**Response**: This argument is saying that if particles are less than 300 nm in diameter, they are not detected. This means that if there are many small particles, then we are undercounting. Once particles are above the detection limit, then we are not sensitive to particle size.

Line 221: "2.5" should be subscripted.

**Response**: Done.

Table 2: Why is the R-squared of the raw data not reported? Reporting the biases across the different methods could also be useful.

**Response**: $R^2$ is now reported in Table 3.

Line 254: The bias of the linear correction being zero indicates that it is being assessed on the same data on which it is calibrated. I'd suggest defining separate calibration and evaluation datasets for each month; see my general comment on this from above.

**Response**: See response to first major comment.

---

## Author Comment (AC2)

Response to Reviewer 2
Retrieval of Bulk Hygroscopic from PurpleAir Sensors, Psotka et al., (egusphere–2024–3618)

**Specific Points**
  **1. Abstract**
  1.1.    Line 3: remove "(empirical)" in favour of just "statistical" or "empirical"

Done.

  1.2.    Line 10: clarify the bias metric used similar to how it was done for MAE

Done.

  1.3.    Line 12: "using our OEM retrieved allowed" - replace retrieved with retrieval?

Done.

  1.4.    Line 13: include the accepted ranges in parenthesis here if possible

Done. The accepted range for atmospheric particulate matter is $0.1 < k < 0.9$ according to Petters and Kreidenweis (2007).

  **2. Introduction**
  2.1. Line 21: add proper citation for [map.purpleair.com](map.purpleair.com) and move to references

Done.

  2.2. Line 22 and 24: It is incorrect to say the PurpleAir monitor makes particulate "measurements" - they estimate the concentrations based on measurements of light scattering and an assumed particle composition

  Agreed, this was re-worded.

  2.3. Line 25: the term "Low-cost sensors" is used but not defined for the reader

  Low-cost changed to PurpleAir.

  2.4. Line 35: Barkjohn et al. expanded on this work in this publication:
    https://www.mdpi.com/1424-8220/22/24/9669

  This reference was added.

  2.5. Line 44: Suggest including https://amt.copernicus.org/articles/15/3315/2022/ as a reference as it compares many of the cited models with sites across Canada/USA

Done.

2.6. Line 45: remove "(also called empirical)" in favour of just "statistical" or "empirical"

Done.

2.7. Suggest adding paragraph(s) describing the OEM method and hygroscopic growth

A reference for OEM was added, and a paragraph about hygroscopicity along with references were added as well.

**3. Methodology**
3.1. What time period is the data from? I believe 2021 based on the comment in this section on data removed for a period in August 2021, but it should be clearly stated what date range was used.
   3.1.1. Suggest adding a study site paragraph to start this section.

A description of the time period of the data and how the data was split by season was added in section 2.2.

3.2. Section 2.1 belongs mostly in the introduction as it is a review of what was done in another study. Lines 75 and 76 should be expanded on here instead with specifics on what was done in this study

We believe this section belongs in the methods because the study referred to, and the equations listed, describe the physical model that was used in our study. Lines were added to make it clearer that this is the method used in our study.

3.3. Line 74: what total is "or about 30%" in reference to?

"or about 30%" was removed.

3.4. Line 79: citation for PurpleAir is missing.

Citation added.

3.5. Line 80: having two sensors is for precision , not accuracy

Done.

3.6. Line 84: clarification is needed whether the A/B comparison was done before or after averaging to daily averages

Done. The comparison was done before the daily averages.

   3.7. Line 86: "typically eliminated about 3% of measurements" - unclear what this 3% applies to, each day?

This is 3% of all raw data measurements before daily averages. This was clarified.

   3.8. Line 90: "due to internal heating and insolation effects "

Done.

   3.9. Line 93: In my experience, the bias in PA RH has a diurnal cycle due to insolation effects on the temperature within the unit. Given this, adding 21% to PA RH may be okay for this dataset but it may not be transferable between sites/time/averaging periods. This should be emphasised in the discussion/conclusion, and I would suggest including justification for the 21% adjustment in the results (ex. a scatter of RH from both monitors before and after)

A scatter of the RH measurements before and after the correction compared to airport values is now presented in Appendix A. In section 3.1.3, it was emphasized that our correction is only valid for our site and time period. In section 4, it is re-iterated that higher quality RH measurements should be used to confirm our method, as our RH correction may not be sufficient.

   3.10. Line 94: "were about 2C high" is vague

This was clarified as about 2 ºC higher than the Airport station values. Given that our method is not sensitive to temperature (section 3.1.2) and we do not correct the temperature reading, we do not need to be more precise.

   3.11. Line 101: link should be cited properly and moved to the reference list

Done.

   3.12. Lines 108 - 121: this paragraph belongs in the introduction

Done.

3.13. Line 108: suggest improving the paragraph transition here. This paragraph should start with the *measurement/monitor* differences, not the *price* differences.

This was re-worded, however, the price is directly related to the differences in the monitors' measurement capabilities.

3.14. Line 111: the plantower sensors independently produce particle counts and concentration estimates using two separate proprietary algorithms (ie. concentrations are not derived from the particle counts). See: https://amt.copernicus.org/articles/13/6343/2020/

Agreed, this was re-worded and references by Ardon-Dryer et al. (2020) and Ouimette et al. (2022) were added.

3.15. Line 117: I don't believe "swelling" is the correct term. The water accretes on the surface of the particle; swelling implies the absorption of water by the particle

This was corrected to an increase in diameter of the particle as is used in literature.

3.16. Line 119: "detect higher concentrations of larger-diameter particles" is incorrect - due to the hygroscopic growth of the particles the sensors detect higher scattering and estimate a higher concentration. The assumed particle density does not change.

Agreed, this was re-worded.

3.17. Lines 123 - 135: belongs in the introduction

We think that this is an important part of the method that we used to implement the physical model to correct the $PM_{2.5}$ measurements, so we will keep it in thee methods section.

3.18. Line 156: what bias metric was used?

The bias metric is now defined in section 2.4.

**4. Results**
  4.1. Line 163: Remove "One model parameter is the particle diameter."

Done.

  4.2. Line 171: define "reasonable ambient range"

Done.

  4.3. Figure 1:
    4.3.1. suggest increasing font size as it is difficult to read even with zooming

4.3.2. Make the x/y axes have the same limits so seasons can be compared visually. As it is now, the Raw fall values look biased higher than that for the winter panel, however the winter axes go out to 60 ug/m3 whereas the fall axes go out to 40 ug/m3

4.3.3. "OEM" and "MLR" need to be defined in the figure caption

Done. Axis limits were made consistent for each season ranging from 0 to 60 ug/m$^3$

4.4. Line 175: what form was the MLR? And what were your coefficients? This needs to be discussed in the methods and results

The MLR is given in Equation 1, this is now referred to explicitly in section 3.2 and the coefficients are reported in Appendix B.

4.5. Lines 177-187: use these sections to describe the results displayed in the figures, not to describe the presentation of the figure. For example, "The raw PurpleAir observations tended to be biased higher than the Ministry PM2.5, which worsens as concentrations increase."

4.6. Line 188: "The physical calibration has a tendency to over-correct at high relative humidity" - is this true? Figure 2 has high and low RH on both sides of the 1:1 line for all seasons. I don't think this is sufficient justification to disregard higher humidity values.

4.7. Figure 2:

4.7.1. Increase font size as it is difficult to read without zooming

4.7.2. The figure caption should state that these are daily observations

4.7.3. The colour scale used makes it difficult to see the mid-range values, this makes the extremes visually stand out. I would suggest binning the RH values into low (<55%) moderate (55- 65) and high (> 65%) and using 3 easily differentiable colours. This would also make it more clear what "higher values of relative humidity" (Line 190) entails.

The results section was entirely restructured and combined with our discussion as we believe this is easier to follow for the reader. The former Figure 2 was removed and replaced with what is now Figure 5 as suggested in comment 4.10.2.

These comments are all addressed with an entirely restructured results section.

4.8. Line 195: the PurpleAir spec sheet lists an accuracy tolerance of 3% ( https://www2.purpleair.com/products/purpleair-pa-ii )

Given how much the humidity differs from our nearby airport site we do not believe that 3% is the true tolerance, so we do not include this value in our paper.

4.9. Figure 3:

4.9.1. Replace "PM2.4" with "PM2.5" for both (a) and (b)

4.9.2. The error bars are not clearly visible on (b) - this should be mentioned in the results section

4.9.3. (b) has "Month" as an x axis label not "Season"

Done.

4.10. Additional recommendations:

4.10.1. A daily mean time series of the observation data would benefit this paper, especially for justifying the bias-adjustment of 21% for the PurpleAir RH.

This was added in Appendix A.

4.10.2. Instead of Figure 2, try RH on the x axis and daily mean bias on the y. That could more clearly show if high RH has an effect on the PurpleAir bias.

This figure was added as Figure 5 in Section 3.3 to show more clearly that bias tends to be negative when relative humidity is above 65%.

**5. Discussion**

5.1. Lines 199-201: this is an important finding that could be highlighted in the abstract

Done.

5.2. Lines 209 - 211: something like this would be great for the results section

Done, this was moved to results.

5.3. Line 217: particle composition/age also varies seasonally as well and is an important factor. The optical properties and hygroscopicity of particulates from a residential wood stove that have not been airborne for long will differ from those from wildfire smoke that travelled from western Canada to eastern Canada. Another factor could be the concentration ranges within each season - concentrations of PM2.5 tend to be lower in Spring/Fall due to less periods of stagnation

The seasonal variation in particulate concentration and composition is now mentioned in Section 3.4.

5.4. Line 218: "biggest" is vague and it is unclear to me what supports this claim

5.5. Line 220: "are more strongly affected" is vague

5.6. Line 228: it is not clear to me how the apparent seasonal differences in overcorrection at high humidities indicates that the physical model does not perform well at high humidities.

These were all re-worded to be more clear. This section is now Section 3.4 so that is where the changes are located.

5.7. Line 231: Table 2 and the summary of it belongs in the results, not discussion

Done.

5.8. Lines 232-233: "The statistically-calibrated data consistently had no bias" - this seems suspect to me (potentially a result of not splitting training/testing data), and is vague. Where was this presented in the results?

The daily bias is now presented in Figure 5, and the average bias is also shown as the horizontal lines in Figure 5. The statistically-calibrated data has an average bias of zero due to the statistical nature of linear regression, this was clarified in the conclusion. The bias would likely be greater than zero if training/testing data was used, but we do not feel that is appropriate for this study.

5.9. Table 2: What about the raw R squared? Did the physical/statistical models improve/worsen the correlation?

The raw $R^2$ is now included in Table 3 in section 3.4. The correlation is improved by both the physical and statistical models.

5.10. Line 236: "greatest" is subjective and was not statistically evaluated - recommend removing this sentence

"The greatest" was changes to "one".

5.11. Line 237: "it is known" should be replaced with "we noticed" and the sentence should be clarified that it is for this site/region - PM2.5 can have dramatic spatial variation

Done.

5.12. Additional limitations
    5.12.1. It is a concern that only one year for a single PurpleAir/Ministry pair was used- are these results transferable to other areas and concentration ranges?

We recommend that other studies are done to investigate our methods are other sites/time periods.

    5.12.2. The data were not split into training and testing datasets, likely overfitting the model and producing overoptimistic performance measurements.

We apologize for the confusion we caused with both reviewers. We should not have said we were deriving a calibration for the measurements. We did not do that. Rather, we are correcting local measurements from a single site to demonstrate the potential for our method to be applied to correct single-day measurements (e.g. on the University of Northern British Columbia Air Quality map (https://cyclone.unbc.ca/aqmap/v2/)), or in the future used to

calibrate networks of PurpleAir sensors as have been done in works like Mailings et al. 2019 and Barkjohn et al. 2021. Then, what we propose above (e.g. a training set) would be the appropriate way to develop a calibration.

The procedure of using a training set for a retrieval is not typically done when solving inverse problems, as the forward model is a complete (as possible) description of the geophysical situation and the instrument function. What is problematic in OEM is how good is the model fit, that is, the value of the cost function. If the cost function is less than 1, then the model is overfitting the measurements (bad!). If the cost function is too high, say (arbitrarily) 10, the model is missing something. A perfect fit of the model to the data would be a value of 1 for the cost function. In our case, the cost values are around 5, meaning we are doing a good job of describing the measurements without overfitting them.

> 5.12.3. The concentration range is fairly moderate, it would be interesting to see the performance during wildfire smoke events where daily mean PM2.5 can exceed 100 ug/m3 (more than twice of what was observed at this site/period)

Unfortunately, the time period we used did not include any wildfire events. We suggest for this method to be tested on different concentrations in the future.

**6. Conclusions**

6.1. Line 258: include the reasonable ranges in parenthesis here if possible

Done.

6.2. Line 260: use the updated url for aqmap ( *https://aqmap.ca/aqmap* ), and move to references with a proper citation

Done.

---

## Author Response (AR2)

**Author Response**

None (no comments given), although 2 minor changes were added to accommodate final suggestions from 1 reviewer.